# Ecomorphological and Age-Related Adaptations in the Tongues of *Phocoena dioptrica* (Spectacled Porpoise) and *Phocoena spinipinnis* (Burmeister’s Porpoise) (Phocoenidae: Cetacea)

**DOI:** 10.3390/ani14233481

**Published:** 2024-12-02

**Authors:** Cleopatra Mara Loza, Carolina Natalia Zanuzzi, Laura Beatriz Andrini, Cecilia Mariana Krmpotic, Alejo Carlos Scarano, Juan Pablo Loureiro, Claudio Gustavo Barbeito, Alfredo Armando Carlini

**Affiliations:** 1CONICET (Consejo Nacional de Investigaciones Científicas y Técnicas), Godoy Cruz 2290 (C1425FQB) CABA, Argentina; carozanuzzi@gmail.com (C.N.Z.); ckrmpotic_pv@fcnym.unlp.edu.ar (C.M.K.); scarano@fcnym.unlp.edu.ar (A.C.S.); acarlini@fcnym.unlp.edu.ar (A.A.C.); 2Laboratorio de Morfología Evolutiva y Desarrollo (MORPHOS), Facultad de Ciencias Naturales y Museo, Universidad Nacional de La Plata, Paseo del Bosque s/n, La Plata 1900, Argentina; 3Laboratorio de Histología y Embriología Descriptiva, Experimental y Comparada (LHYEDEC), Facultad de Ciencias Veterinarias, Universidad Nacional de La Plata, La Plata 1900, Argentina; 4Cátedra de Citología, Histología y Embriología, Facultad de Ciencias Médicas, Universidad Nacional de La Plata, La Plata 1900, Argentina; landrini69@gmail.com; 5Departamento de Ambiente y Turismo, Universidad Nacional de Avellaneda, Buenos Aires B1870, Argentina; 6Fundación Mundo Marino, San Clemente del Tuyú, Buenos Aires B7105, Argentina; jploureiro@gmail.com

**Keywords:** tongue, ontogenetic differences, countercurrent, thermoregulation, odontoceti

## Abstract

The tongues of vertebrates reflect their adaptations to various feeding strategies, the types of food they consume, and the environments they inhabit. In cetacean, the macro- and microanatomical aspects of the tongues of few species have been studied. Here, we analyzed, for the first time, the morphology of the tongues of two porpoise species (Spectacled and Burmeister’s Porpoises; juvenile and adults), whose biology is little known. We describe a range of novel aspects related to ontogenetic morphological differences and document the finding of thermoregulatory structures and components of the immune system. Differences between juvenile and adult individuals of the same species, as well as between juveniles and adults of both species, may be related to their feeding types and/or geographical distribution. Additionally, we found a lingual vascular system, which has only been mentioned for baleen whales and the sperm whale (but never for smaller-toothed cetaceans), that may participate in the thermoregulation of these individuals. Both species have marginal papillae, but only in Burmeister’s Porpoise were small taste buds, possibly vestigial, found. A better understanding of the biology of these two porpoise species will help to develop useful strategies that contribute to their protection in the near future.

## 1. Introduction

The tongues of vertebrates exhibit structural variations that reflect adaptations to diverse feeding strategies, to different types of consumed food items, and the environment that they inhabit. These variations are evident in those differences observed among terrestrial mammals, amphibians, and exclusively aquatic species [1,2]. The tongue has been studied in numerous taxa of terrestrial mammals, particularly in domestic species [3], but also in other groups such as armadillos and non-domestic carnivores, e.g., [4,5,6,7,8,9,10,11,12]. In general, most mammals show the dorsal surface of the organ characterized by the presence of projections from the mucosa, covered by an epithelium with varying thicknesses and degrees of keratinization, which together form mechanical papillae (filiform, conical, lenticular, and marginal), and eventually gustatory papillae containing taste buds (fungiform, circumvallate, and foliate) [13]. The mechanical papillae are associated with the capture, retention, compression, and partial erosion of food items, and possibly the selection of retained food items (filtration) [1,14]. In turn, the taste buds enable the perception of chemical substances, both from the environment and from the food itself. The distribution, shape, size, and number of mechanical and gustatory papillae vary among different clades and species of mammals, both terrestrial and marine [1]. Thus, the type, shape, relative quantity, and distribution of lingual papillae (mechanical and with taste buds) have frequently been linked to different feeding habits and the environments in which they live [7]. Among the most common functions of the tongue in most mammals are grasping and transporting food into the oral cavity, as well as manipulation and positioning during the masticatory cycle and swallowing. In addition, mammal tongues also participate in acquiring liquids, licking, sucking during nursing, perceiving chemical signals and food textures, helping to vocalize, coughing, regurgitating, and eventually thermoregulation [15].

Regarding morphological studies on tongues of non-cetacean marine mammals, research has been partially conducted on mustelids, such as *Enhydra lutris* [16], various species of Otariidae, Phocidae, and Odobenidae [8,14,17,18,19], and sirenians [20,21]. The tongues in these animal groups show morphofunctional differences that were associated with diet type, foraging habits, and dental morphologies, which in turn are linked to the foods they consume [14,22,23,24]. Concerning cetaceans, ecomorphofunctional studies have been conducted on the tongues of some species, such as *Stenella coeruleoalba* [25], *Mesoplodon stejnegeri* [26], *Pontoporia blainvillei* [27], *Phocoena phocoena*, and *Tursiops truncatus* [17,28], focusing on both anatomical and histomorphological aspects.

Considering the relevance of the tongue, and that little is known about that organ in *Phocoena dioptrica* and *Phocoena spinipinnis*, the objective of our contribution is to describe the macro- and microscopic morphology of the tongue in these two porpoise species (in juveniles and adults) and try to interpret them in an ecomorphological context. Therefore, the external morphology, the presence, diversity, and distribution of mechanical papillae, the thickness of its lining epithelium and the stratum corneum, the existence and distribution of taste buds, the vascularization, and the organization of the musculature were compared and analyzed adaptively in two species of austral porpoises (*P. dioptrica* and *P. spinipinnis*) and at two ontogenetic stages. Finally, the data were analyzed in relation to the known preferential feeding habits of each species and their geographical distribution.

## 2. Materials and Methods

The specimens studied here were found dead on the shores of San Clemente del Tuyú, Buenos Aires, Argentina (longitude: W56°43′24.64″; latitude: S36°21′24.98″), in July 2021. One adult female (M6020) and a juvenile male (M6121) of *Phocoena dioptrica* and also two females, one adult (M8522) and one juvenile (M12921), of *Phocoena spinipinnis* were analyzed. All specimens were housed at the Centro de Rescate y Rehabilitación of Fundación Mundo Marino collection; the dissections were carried out in a necropsy room by veterinarians. The Institutional Committee for the Care and Use of Laboratory Animals (CICUAL, Veterinary School, National University of La Plata) approved all the procedures (protocol N°231009-4). The tongues were fixed in 10% formalin with seawater (to maintain the osmolarity of the soft tissues); subsequently, they were photographed with a digital camera for the gross anatomy, and observed and photographed with a Nikon SMZ 645 stereoscopic microscope for details. One longitudinal half of each tongue was processed for examination using scanning electron microscopy in a FEI SEM microscope at CINDECA, Facultad de Ciencias Exactas, Universidad Nacional de La Plata. The other half was preserved in 70% ethanol until further processing for optical microscopy studies. Following previously established protocols, the samples were dehydrated through increasing concentrations of ethanol from 70% to 100%, embedded in paraffin, and sectioned into histological slices of 3 µm to 5 µm in thickness. The sections were stained with hematoxylin and eosin (H&E), orcein (to identify elastic fibers), and Gomori’s trichrome stain (to differentiate muscle fibers from collagen fibers) [29]. For the microscopic studies, the different regions were analyzed (vertex, anterior body, posterior body, transition zone, and root), previously defined based on macroscopic anatomical features. The vertex was determined by the presence of marginal papillae; the body was subdivided into anterior and posterior based on the presence of these papillae and the characteristics of its surface (smooth); the transition region was identified by the presence of small openings forming a “V” shape, pointed distally; and, finally, the root was defined as the most caudal sector with a rough surface. Each of these regions was further subdivided into medial (R1, R3, R5, and R7) and lateral (R2, R4, R6, and R8) regions (Figure 1).

### 2.1. Age of the Specimens

*Phocoena dioptrica* adult males are larger than adult females and they possess a distinctive dorsal fin that is bigger than that of females. Thus, the specimen M6121, a male measuring 143 cm, with a small dorsal fin (similar to that of females) and marginal papillae in the tongue [30], was classified as a juvenile. In contrast, the specimen M6020, a pregnant female measuring 181 cm in length, was considered an adult.

The specimen *P. spinipinnis* M8522, a female weighing approximately 150 kg, was regarded as an adult. Meanwhile, the female specimen M12921 was classified as a neonate, since it measured 100 cm, a length which is approximately the normal size for newborns in these species of porpoises [31,32,33].

### 2.2. Immunohistochemistry

To confirm the presence of lamellar corpuscles in the studied areas of the tongues, immunohistochemistry was performed, following previous protocols [34]. Briefly, paraffin-embedded sections were dewaxed, rehydrated, and then incubated with 0.03% H_2_O_2_ in methanol (purity P99.0%) for 30 min at room temperature (RT) to inhibit endogenous peroxidase activity. Then, antigen retrieval was performed in citrate buffer (0.01 M, pH 6.0) and microwaved at 800 W for 1 min. Non-specific binding was blocked with bovine serum albumin 1% in PBS 0.2 M for 30 min, and then slides were incubated at 4 °C overnight with polyclonal rabbit anti-S100 antibody (Dako, North America, Inc., Carpinteria, CA, USA). After washing in PBS, slides were incubated with ImmPRESS^®^ HRP Universal Antibody (Horse Anti-Mouse/Rabbit IgG) Polymer Detection Kit, Peroxidase (MP-7500) (Vector Laboratories, Inc., Burlingame, CA, USA), for 30 min at RT. Then, slides were rinsed three times in PBS for 5 min each. Liquid 3,3′ diaminobenzidine tetrahydrochloride (DAB) was used as the chromogen (DakoCytomation, Carpinteria, CA, USA).

### 2.3. Quantitative Analysis

Digital photographs were taken of histological sections from the eight regions of the tongues of each specimen. Photos were taken using a Leica DM500 microscope and a Leica IC500W digital camera through LAZ 4.0 software. The images were obtained with a scale for calibration purposes. Subsequently, using ImageJ 1.50 software, 10 measurements of the epithelial thickness were taken from each region of the tongue in each individual. This allows the variability in epithelial thickness to be captured and provides more accurate central tendency measurements (Appendix A). To visualize the variation in epithelial micromorphology in each individual, the shape of the epithelial ridges from each zone was evaluated at 500 µm intervals using tpsDIG2 (v. 2.32) software. All graphs were created using R v4.2 software.

## 3. Results

The samples from the four specimens were studied and compared taking into account both a macroscopic perspective (examining their general and detailed structure using SEM) and a microscopic perspective (histological structure).

### 3.1. General Macroscopic Structure

The tongues of both species show differences in the quantity and distribution of marginal papillae at their vertex, not only among juveniles and adults, but also between juveniles. The dorsal surface is smooth in the body region and shows variable roughness in the root area (Figure 2).

#### 3.1.1. *Phocoena dioptrica* (M6021—Adult Female)

The tongue outline is sub-triangular with a more acute tip than in *P. spinipinnis*; the dorsal surface of the vertex (R1 medial and R2 lateral) is completely smooth. However, its free edges have marginal papillae. The more anterior papillae are broad-based and spatula-shaped, while the more posterior ones are digitiform, with a narrower and elongated base (Figure 3).

The dorsal surface of the anterior body region (R3 medial and R4 lateral) is smooth, while its margins have a series of thickenings with visible pores, which were not found on the sides of the posterior body region (R5 medial and R6 lateral) (Figure 3). However, in this latter sector, lenticular or verrucous structures were observed to be unevenly distributed on the surface (Figure 3). The transition zone exhibits intermediate characteristics between those described for the body and the root.

The root (R7 medial and R8 lateral), unlike the other described areas, has a rough surface with no papillae or other evident structures (Figure 3).

#### 3.1.2. *Phocoena dioptrica* (M6121—Juvenile Male)

The general characteristics of the tongue in this specimen are similar to those of the adult specimen, with the difference that its tip has a rounded contour, and that in (R2), the marginal papillae are more numerous (Figure 4). The anterior (R4) and posterior (R6) body regions exhibited the same characteristics as in the adult specimen, but the marginal zone of both regions has more pronounced elevations or thickenings compared to the adult (Figure 4). In the transition region, a series of depressions were observed, aligned in a bilateral convergent row, forming a ‘V’ shape pointing distally. The root also has roughness, but it is not as deep and extensive as in the adult specimen (Figure 4).

#### 3.1.3. *Phocoena spinipinnis* (M8522—Adult Female)

The tongue of this specimen has a rounded distal contour, compared to the adult of *P. dioptrica*, and a rougher surface. The vertex (R2) has very poorly developed marginal papillae, even less than in the adult specimen of *P. dioptrica*. In the anterior body (R4), there are thickened margins with similar characteristics to those of the adult specimen of *P. dioptrica*. The surface of the root is more irregular and highly verrucous, with numerous folds, compared with the other specimens (Figure 5).

#### 3.1.4. *Phocoena spinipinnis* (M12921—Juvenile Female)

Unlike the juvenile of *P. dioptrica*, *Phocoena spinipinnis* has a greater number of marginal papillae, with more varied shapes and sizes, and even arranged in double or triple rows along the edges, both at the vertex (R2) and the body (Figure 6). In the anterior body (R3, R4), the dorsal surface is completely smooth, except for the presence of thickenings on its edges, whereas in the posterior body region and towards the medial one (R5), the surface has protruding structures with marked reliefs, resembling warts, arranged in rows, forming a ‘V’ shape between the body and the root (Figure 6), also including the transition zone. The surface of the root is rough, and there are numerous foliated papillae on the sides (Figure 6).

### 3.2. Scanning Electron Microscopy (SEM)

#### 3.2.1. *Phocoena dioptrica* (M6021—Female Adult)

The dorsal surface of the vertex (R1 and R2) is smooth, with a series of small marginal papillae at its distal end, becoming bigger and in larger numbers towards the proximal end. The size and shape of these papillae are variable; some are small lateral elevations (Figure 7A), while others are larger, lobulated, and some even bifurcated (Figure 7B). In the medial region of the anterior body (R3), the dorsal surface is smooth with localized depressions (ranging from shallow to deep) of variable sizes and shapes (e.g., circular to oval), and with a random distribution (Figure 7C). In the posterior body (R5 and R6), the dorsal surface is slightly rougher, with fewer depressions that are predominantly oval-shaped (white arrows in Figure 7D).

The root (R7 and R8) has an even rougher surface with crypts and some depressions similar to those of the body, but in smaller quantities and with a random distribution (Figure 7E,F).

#### 3.2.2. *Phocoena dioptrica* (M6121—Juvenile Male)

The marginal papillae on the vertex (R2) are more numerous, larger, and more variable in shape (rounded, pyramidal, columnar of different heights) compared to those of the adult specimen. Some are bilobulated or even cordiform in their outlines (Figure 8A,B). The body has a primarily smooth dorsal surface, with few isolated depressions. The margins of the anterior and posterior body (R4, R6) are smooth, with few depressions (white arrow in Figure 8C,D).

The root (R7, R8) features a series of interrupted grooves close to the boundary with the posterior body, and with variable orientations (Figure 8E,F).

#### 3.2.3. *Phocoena spinipinnis* (M8522—Adult Female)

Only a few marginal papillae are observed on the vertex (R2), while the anterior surface exhibits roughness with semicircular contour reliefs (Figure 9A,B), and in the posterior region, the vertex has a smooth surface with only slight lateral reliefs. The anterior body (R3 and R4) features a wide variety of surface irregularities, reliefs, and flattened papillae with circular or triangular outlines, some of which are conical. The posterior body, on its medial region (R5), has extensive crypts and depressions, whereas laterally (R6), it shows a predominantly smooth surface (Figure 9C,D). In the transition zone and root, the crypts deepen and increase in complexity (Figure 9E,F).

#### 3.2.4. *Phocoena spinipinnis* (M12921—Juvenile Female)

The marginal papillae on the vertex (R1 and R2) are variable in size and shape, with the more posterior ones being longer and more numerous (Figure 10A–C).

On the body (R3 and R4), there are some depressions, similar to those previously described, but with shallower depressions arranged radially and heterogeneously distributed (white arrows in Figure 10D,E).

In the transition zone, it was determined that the groove (previously described in the macroscopic examination) corresponds to a series of crypts distributed in various directions and with varying depths. These series of crypts form the diagonal lines of the “V” that delineates the boundary with the posterior body (Figure 10F). On the sides of the root, numerous papillae with varied contours (lenticular, elongated, rounded) were observed, as well as several small depressions scattered across the predominantly smooth lingual surface (white arrows in Figure 11).

### 3.3. Microscopic Structure and Histology

Below is a general histological description of the tongues in porpoises.

The tongue is a muscular organ covered by a mucosa. The mucosa is formed by a cornified and parakeratotic stratified squamous epithelium and a lamina propria of loose connective tissue, richly vascularized and innervated, with elastic fibers and isolated skeletal striated muscle fibers. Additionally, mucous alveolar or acinar glands are present. The center of the organ consisted principally of skeletal striated muscle fibers arranged in horizontal (lateromedial), vertical (dorsoventral), and longitudinal (craniocaudal) directions. In addition, variable amounts of unilocular adipose tissue and vascular adaptations (periarterial venous retia) are present at specific areas.

#### 3.3.1. *Phocoena dioptrica* (M6021—Adult Female)

The dorsal and ventral surfaces of the medial vertex (R1) are lined by a cornified parakeratotic stratified squamous epithelium. The lamina propria forms irregular projections of loose connective tissue at the subepithelial level, but more deeply, it becomes denser, with elastic fibers, blood vessels of variable caliber, and interspersed skeletal striated muscle fibers (Figure 12A,B). In contrast, the connective tissue on the ventral surface is looser, with numerous larger blood vessels, lymphatic vessels, and a higher density of nerve fibers than the dorsal side. Between the mucosa and muscle, a layer of dense connective tissue with abundant elastic fibers and clusters of unilocular adipose cells is apparent (Figure 12C). The connective tissue surrounding the bundles of muscle fibers also contains elastic fibers and unilocular adipocytes.

The marginal papillae of the vertex (R2) are composed of cornified parakeratotic stratified squamous epithelium and a lamina propria of loose connective tissue with numerous lymphatic vessels, sparse and thin collagenous and elastic fibers, and isolated striated muscle fibers. At the dorsal surface, the subepithelial connective tissue is loose and then it becomes dense, with a higher proportion of collagenous and elastic fibers.

The muscle exhibits differences in the organization. The most superficial bundles are arranged in three spatial planes: horizontal (medial–lateral direction), vertical (dorsal–ventral direction), and longitudinal (craniocaudal direction). More deeply, longitudinal bundles predominate.

In the anterior body (R3 and R4), the epithelium shows a greater number of strata than that of the vertex. The connective tissue has similar characteristics to those described in the vertex, and more caudally, it sends higher number of projections, with isolated muscle fibers. On the lateral surface of the body, the epithelium forms foliated-like structures with multiple connective tissue cores similar to the foliated papillae found in other mammals, but with no taste buds. Towards to the posterior body, glands begin to appear.

Throughout the body, periarterial venous retia (a central muscular arteria and two or more peripheral veins) are found. The associated connective tissue of these vascular structures shows abundant elastic fibers (Figure 12D).

In the anterior body, the musculature is arranged similar to that described at the vertex but with few horizontal bundles. In contrast, the posterior body shows longitudinal (craniocaudal) and vertical (dorsoventral) bundles at the surface, and more deeply, the longitudinal bundles predominate, being scarce in the horizontal arrangement.

In the transition region, the epithelial folds are more separated from each other and show several cores of loose connective tissue and few elastic fibers but are also without associated taste buds. Additionally, alveolar glands, lined by columnar and mucous PAS-positive cells are well developed, and accompanied by connective tissue enriched in elastic fibers (Figure 12E). The muscular bundles are arranged in four alternated directions from the surface: vertical, horizontal, vertical, and longitudinal. Furthermore, deep in the muscle, periarterial venous retia are also found. The connective tissue surrounding the muscle fiber bundles also contains elastic fibers.

The root region exhibits similar characteristics throughout its extent (Figure 12F). The mucosa forms depressions (foveae) which continue with folds of epithelial tissue at the bottom of which glandular ducts open. The glands are well developed, predominantly alveolar, and lined by mucous-secreting cells. The accompanying connective tissue contains abundant blood vessels. Periarterial venous retia are also present. The musculature follows a vertical pattern, even among the glands, and then it shows a horizontal arrangement.

In the connective tissue of the mucosa of the vertex, body, transition, and root regions, positive s-100 lamellar corpuscles (axon terminals surrounded by Schwann cells, which build lamellar structures in the cytoplasmic process) are present (Figure 13A,B).

#### 3.3.2. *Phocoena dioptrica* (M6121—Juvenile Male)

The tongue of this specimen exhibits several similarities with that of the adult female. The mucosa of the anterior and lateral vertex (R2) features marginal papillae (Figure 14A). In both regions, it is covered by the same previously described epithelium. The lamina propria sends numerous projections towards the epithelial tissue, and it consists of loose connective tissue with scant isolated striated muscle fibers. The dorsal surface is smooth, with loose connective tissue that gradually becomes denser, with abundant blood vessels of variable calibers. The mucosa of the lateral and ventral regions forms epithelial folds, with more extensive and looser connective tissue than that found at the dorsal side and, also, than the one described in the adult specimen; elastic and collagen fibers and lymphatic vessels are also found. The connective tissue becomes denser in the muscular tunica with numerous groups of unilocular adipocytes, which are more abundant than in the tongue of the adult specimen. Regarding the musculature, in (R1), it is organized into vertical and longitudinal bundles, lacking those horizontal bundles present in adults. In (R2), the musculature shows a plexiform pattern.

In the lateral and ventral sector of the anterior body (R4), epithelial folds were observed with several internal cores of loose connective tissue. These folds become more organized in the most caudal sector of this region. Between the connective tissue of the lamina propria and the muscular tunica, there is a dense connective tissue layer with abundant elastic fibers (Figure 14B). Periarterial venous retia, similar to those observed in the adult specimen, are also present (Figure 14C). The posterior body region (R5–R6) does not exhibit differences, except for the presence of latero-ventral epithelial glands and folds.

The transition region exhibits the same characteristics described for the adult specimen, although the epithelial folds are less pronounced and glands are more abundant (Figure 14D). In the most caudal sector, glandular ducts with exfoliated cells in their lumen and associated periductal lymphoid tissue are present. Periarterial venous retia and depressions (fossae) with internal folds of the mucosa are found. Musculature generally follows the body’s pattern, although it tends to be less organized at the dorsal level.

The root (R7–R8) exhibits, throughout its length, similar characteristics to that described for the tongue of the adult specimen (Figure 14E). The epithelial mucosal folds are more pronounced, and at their bases, the ducts of mucous alveolar glands open. These glands reach the depth of the muscle. In addition, diffuse lymphoid tissue associated with glandular ducts became more evident, and nodular lymphoid tissue surrounding small-caliber blood vessels is also observed (Figure 14F). Unilocular adipose tissue is more developed in comparison to other regions and the adult specimen. The periarterial venous retia described in other regions are also evident.

In the connective tissue of the mucosa of the vertex, body, transition, and root, positive s-100 lamellar corpuscles are present, as in the adult specimen.

#### 3.3.3. *Phocoena spinipinnis* (M8522—Adult)

The adult specimen shows marginal papillae extending to the posterior lateral vertex (R2), lined by the same epithelium described in the tongue of *P. dioptrica*. The dorsal surface is smooth, but with highly vascularized loose connective tissue projections beneath the epithelium. In addition, unilocular adipose tissue appears between the mucosa and musculature and disorganized the musculature arrangement, being predominantly longitudinal (Figure 15A,B).

In the dorsal surface of the anterior body (R3 and R4), epithelial folds with highly vascularized connective tissue projections become evident. In this region an epithelial structure morphologically similar to a taste bud was found (Figure 13C). The mucosal folds are also found in the posterior body (R6), and acinous–tubular mucous glands are present (Figure 15C). In contrast to *P. dioptrica*, periarterial venous retia are not observed. The musculature arrangement alternates between vertical and longitudinal bundles. Then, it turns longitudinal.

The transition region shows abundant glands and conic-like epithelial folds. Sparse groups of unilocular adipocytes are also present (Figure 15D).

The root, throughout its length, features numerous mucosal folds with a significant number of connective tissue indentations and extensive development of mucous acinar glands that reach the muscular tunica. Abundant elastic fibers, nerves, and ganglia are also present in the surrounding connective tissue. Adipose tissue becomes more evident among the muscular bundles, which predominantly show vertical orientation (Figure 15E,F).

As described in the tongues of *P. dioptrica*, in the connective tissue of the mucosa coat of the vertex, body, transition and root, positive s-100 lamellar corpuscles are also observed.

#### 3.3.4. *Phocoena spinipinnis* (M12921—Juvenile)

The characteristics of the mucosa of the anterior and lateral vertex (R2) features higher number of marginal papillae with more variable shapes and sizes compared to the juvenile specimen of *P. dioptrica* (Figure 16A). The connective tissue is loose and sends small vascularized indentations with abundant elastic fibers. Towards the medial region, less developed epithelial folds are observed. As in the adult specimen, there are abundant collagen fibers between the bundles of muscular fibers. The musculature arrangement alternates between longitudinal and vertical bundles, as described in the tongue of the juvenile specimen of *P. dioptrica*. In the transition region, an epithelial structure that morphologically could correspond to a taste bud was found (Figure 13D).

In the anterior body regions (R3, R4), the epithelial tissue is smooth, but small projections of highly vascularized connective tissue are identified (Figure 16B). Towards the more caudal and lateral part of this region, the surface becomes more irregular due to the presence of some papillae with loose connective tissue and delicate elastic fibers that become denser towards the muscle tissue. The musculature is not arranged in as orderly a way as in the vertex.

The posterior body (R5, R6) and the transition regions show the same general histological characteristics, although the mucosa folds and forms papillae with a conic-like shape (Figure 16C–E). Mucous-secreting acinar glands are also found. Unlike the adult specimen, periarterial venous retia are identified. The anterior root region shows mucosal folds with pronounced grooves, abundant glands, nerves, and ganglia.

In the posterior root, the folds, grooves, and glands are more numerous (Figure 16F). Caudally, similar to the juvenile specimen of *P. phocoena*, areas of lymphoid tissue associated with glandular ducts are observed. Periarterial venous retia are also found (Figure 16D).

In the connective tissue of the mucosa of the vertex, body, transition, and root, positive s-100 lamellar corpuscles are present, as in the adult specimen.

### 3.4. Quantitative Results

The distribution of the epithelial thickness of each region of the tongue of each animal is shown in Figure 17. A common pattern of epithelial thickness variation is observed among the regions. In general, the epithelial thickness increases from the root to the vertex (see Appendix A). Despite this, in the juvenile specimen of *P. spinipinnis*, the thickness remains relatively constant. The transition shows higher variability in the epithelium’s thickness. This aspect is not observed in the juvenile specimen of *P. spinipinnis*.

In particular, when comparing the values of both ontogenic stages, adult and juvenile, of *P. dioptrica* (M6021 and M6121), the transition region’s epithelial thickness increases as it approaches the most cranial regions (posterior and anterior body and vertex). In contrast, juvenile *P. spinipinnis* shows a thicker epithelium than that of the adult specimen in the caudal regions (anterior and posterior root and transition), which then become thinner in the more cranial regions (anterior body and vertex).

#### Morphology of the Epithelial Ridges

Epithelial ridges, formed at the interface between the epithelium and the underlying connective tissue, exhibit irregular morphology and vary in depth (Figure 18). The most pronounced ridges are observed in juvenile and adult *P. dioptrica* specimens. The areas of greatest interdigitation generally occur at the vertex (Figure 18).

## 4. Discussion

The general morphological characteristics of the tongues of porpoises align with the general pattern described in other odontocete cetaceans, featuring primarily smooth dorsal surfaces [25,26,27]. However, the dorsal surface of the adult *P. spinipinnis* has a rougher texture compared to those of the other analyzed specimens. These texture differences (although no other mechanical papillae apart from the marginal ones were observed with SEM) might be associated with its diet, which includes a greater variety of food items (e.g., pelagic and demersal fish such as *Engraulis* sp., *Anchovia* sp., *Merluccius* sp., *Sardinops* sp., *Odontesthes* sp., *Normanichthys* sp., *Sparus pagrus, Clupea* sp., *Cynoscion guatupuca,* squids like *Loligo* sp., and mysid crustaceans and krill shrimp [35]) than *P. dioptrica.* Although the diet of *P. spinipinnis* varies by geographic region, it consistently includes a broader range of items in comparison with *P. dioptrica*. In contrast, the tongue of the adult female of *P. dioptrica* has a smooth texture (similar to the juvenile specimens of both species), which may relate to a less diverse diet consisting of smoothing surface prey (e.g., *Engraulis* sp., some stomatopods, tunicates, cephalopods—[30,36]).

In the dorsal view, marginal papillae, varying in shape and size, are more abundant in lactating specimens; with age, these papillae become less significant as the calf matures [17]. In this regard, Kastelein [17] described the presence of more marginal papillae in a 1-month-old *P. phocoena* specimen compared to a 6-month-old one. Both juveniles studied here exhibit differences not only in shape but also in the number of papillae, which could be also associated with their estimated different ages or to the type and composition of maternal milk. It is known that cetacean milk is typically very thick due to its high fat content, which can reach up to 50%, and is virtually semi-solid in cold conditions, like the ocean open waters [37]. A milk with lower fat content would be more fluid and thus a greater number of marginal papillae may be needed to ensure an effective sealing with the nipple and prevent milk loss and/or contamination with seawater. Alternatively, this could be associated with the duration of milk suction (the shorter the duration, the more effective and rapid the suction should be), or with the timing of nursing (whether the animals are swimming or stationary); both situations might require their offspring to develop a more effective “anchoring” to the nipple.

Regarding this, there are limited data on the milk composition and nursing behavior of the two species studied here. However, in *Phocoena phocoena,* the lactation is very short compared to other odontocete cetaceans (solid food intake begins at two months old and weaning occurs at 8–12 months) [38,39], and it also has a milk lipid percentage close to 45.8% [38]. In a recent contribution, Denuncio et al. [40] reported a much lower lipid concentration in the milk of *P. spinipinnis*; a more fluid milk secretion in *P. spinipinnis* might require a better seal between the offspring oral cavity and the nipple, which could support the macro- and microscopic characteristics described here.

As previously mentioned, histological and morphometric studies also show differences between species, ontogenetic stages, and along homologous regions of the tongues. Concerning epithelial thickness, it is greater at the vertex and decreases towards the lingual base in both adult specimens and the juvenile of *P. dioptrica*; however, in *P. spinipinnis*, the opposite occurs, with the greatest thickness found at the base (Figure 19). Correspondingly, the same pattern is observed with the projections of the underlying connective tissue (it is more pronounced at the vertex in all specimens, except for the juvenile of *P. spinipinnis*). Both characteristics (deeper thicknesses and interdigitations) are possibly associated with areas of the tongue exposed to higher friction, as is interpreted for cetacean skin [41].

The inverted pattern observed in the juvenile specimen of *P. spinipinnis*, along with the greater number of marginal papillae, may suggest that a higher grip area is required at the apex and that increased friction during lactation is more pronounced towards the base. These speculations can only be confirmed with further studies on the biology of these species.

Concerning taste buds, in this study, we only found them in the tongues of *P. spinipinnis* specimens. Although behavioral evidence suggests that gustation is very limited in cetaceans [42], many studies have shown that in these animals, taste buds are few and involution with age [43]. Furthermore, in several aquatic animals that lack taste buds, the presence of solitary chemosensory cells expressing the same taste receptors as their neuroepithelial counterparts in taste buds has been suggested. These cells respond to diverse molecules, primarily produced by microorganisms, and trigger defense responses in collaboration with the immune system [43,44]. Although it was not an aim of this current work to study the existence of this cell population, it will be interesting to know whether the absence or scarcity of taste buds in the tongues of porpoises is compensated by the presence of solitary chemosensory cells, and whether these cells also participate in local defense mechanisms. Regarding mechanoreceptors, we found lamellar corpuscles in the tongues of both species, as reported in *P. phocoena* [28]. Both species have mucous secretion glands located in the posterior body region, transition zone, and lingual base, and considering that the mucus is immiscible in hypertonic saline water, it could act as a film that prevent dehydration (especially in the zones were the epithelium is thinner). No differences were observed in the distribution and abundance of these glands, neither between adults and juveniles nor between species. The lack of a serous gland may be associated with the few taste buds present.

The intrinsic dorsal musculature of the tongue (but not the gular or hyoid apparatus musculature [45,46,47]) is arranged in horizontal, vertical, and longitudinal bundles. These particular arrangements are associated with tongue compression (bilateral shortening by horizontal muscle bundles), depression (flattening by vertical muscle bundles), and antero-posterior shortening (by longitudinal bundles). Additionally, the orderly alternation and crossing of these bundles would facilitate a more coordinated and continuous prey and milk suction.

Primarily at the tongue’s vertex, craniocaudal or antero-posterior fibers (green fibers in Figure 20) predominate, and their contraction would generate shortening movements. These bundles are evenly distributed in the adult and juvenile *P. dioptrica* and in the juvenile *P. spinipinnis*. In contrast, they are preferentially laterally relocated in the adult *P. spinipinnis*. In all cases, this arrangement would assist not only in nipple drainage but also in food incorporation by suction (a mechanism described in the common porpoise [17]). However, the musculature in the adult *P. spinipinnis* shows a plexiform distribution in the anterior body, which may be related to the more varied diet and, consequently, to the need for more diverse and plastic movements.

Dorso-ventral-oriented fibers generate crushing movements when they shorten. While these fibers are consistently present at the different regions of the tongue and among different species and ages, they are most predominant in the transition region, which is the main area involved in moving food backward and in swallowing.

Adipose tissue is abundant and even interspersed among the muscle bundles in the tongue of the adult *P. spinipinnis*. This might prevent temperature loss and serve as a potential energy reserve, as occurs in other cetacean species, such as large whales. In these animals, the high proportion of adipose tissue serves as thermal insulation, and seasonal fat reserves linked to prolonged fasting [46].

The periarterial venous retia found appear to fit with the countercurrent systems described in the cetacean skin [48] and in the tongues of some mysticetes [15]. Additionally, Werth [49] mentions the existence of countercurrent vascular systems in the tongue of the sperm whale (*Physeter macrocephalus*), which may function to prevent excessive temperature loss due to constant oral cavity exposure to cold waters. Here, we report for the first time the existence of these countercurrent systems in the studied species. They are present in all the studied regions of the tongue of adult *P. dioptrica*, and from the anterior body to the root of the tongues in the juvenile stages of both species. No periarterial venous retia were found in the adult specimen of *P. spinipinnis* (Figure 21 and Figure 22). Similarly to those in the skin, these mechanisms would prevent heat loss at the oral cavity. The differences observed between species may be related to their geographic distribution. *P. dioptrica* ranges from southern Brazil to Tierra del Fuego, the Falkland Islands, South Georgia, Kerguelen, and circumpolar oceanic waters, usually cold waters below 10 °C, whereas *P. spinipinnis* is associated with the coastal waters of South America, from northern Peru to Cape Horn and up to the south of Brazil, along the Atlantic coast [35]. Thus, the presence of a countercurrent system throughout the tongue in *P. dioptrica* might be explained by the need for highly effective thermal isolation in cold waters. Their identification in the juvenile specimen of *P. spinipinnis* alone could be understood as a transient retention of insulating function during this life stage, than then is replaced by adipose tissue in the adults, which is in line with their distribution in warmer waters. Given the limited knowledge about the biology, reproductive behavior, and feeding pathways of both *Phocoena* species studied here, thermal insulation (through countercurrent systems or fat accumulation) related to water temperatures, or energy reserves for potential fasting periods, are both plausible explanations.

Regarding the immune system in cetaceans, primary and secondary lymphoid organs and mucosa-associated lymphoid tissue (MALT) are similar to those of terrestrial mammals [50]. However, some lymphoid structures, such as lymphoepithelial aggregates in the larynx and anal canal, are unique to cetaceans and may ensure an efficient immune response to constant antigenic exposure. Although macroscopic and microscopic descriptions of lymphoid tissue and organs have been published for both odontocetes and mysticetes [25,50,51,52], none so far have mentioned or described the presence of diffuse and/or nodular lymphoid tissue in the tongues of the species studied here. In this study, we found them in the juvenile *P. dioptrica* and *P. spinipinnis* specimens. To date, there is no study on odontocetes that confirms the existence of tonsil-like structures in the tongue [27,50]. Only Yamasaki et al. [27] described a few lymphoid nodules in the tongues of *Pontoporia* sp., *Platanista* sp., and *Inia* sp., but did not define them as true lingual tonsils, as found in other mammals. Comparatively, the distribution of the lymphoid tissue found in juvenile *P. dioptrica* and *P. spinipinnis* specimens resembles not only the one described in small ruminants but also that found in the anal tonsils of *Tursiops truncatus* [51]. The absence of such lingual tonsils in adult *P. dioptrica* and *P. spinipinnis* may be understood as age-related involutional changes, as reported for the laryngeal and anal tonsils of *T. truncatus* [53].

## 5. Conclusions

The general morphology of the tongues of the two porpoise species studied in this paper aligns with that of other cetaceans previously analyzed. However, we document a range of novel aspects related to ontogenetic morphological differences, a lingual countercurrent vascular system not previously reported in small cetaceans, and lymphoid aggregates. Both species have mechanical marginal papillae, and in *P. spinipinnis,* small and likely vestigial taste buds were observed.

## Figures and Tables

**Figure 1 animals-14-03481-f001:**
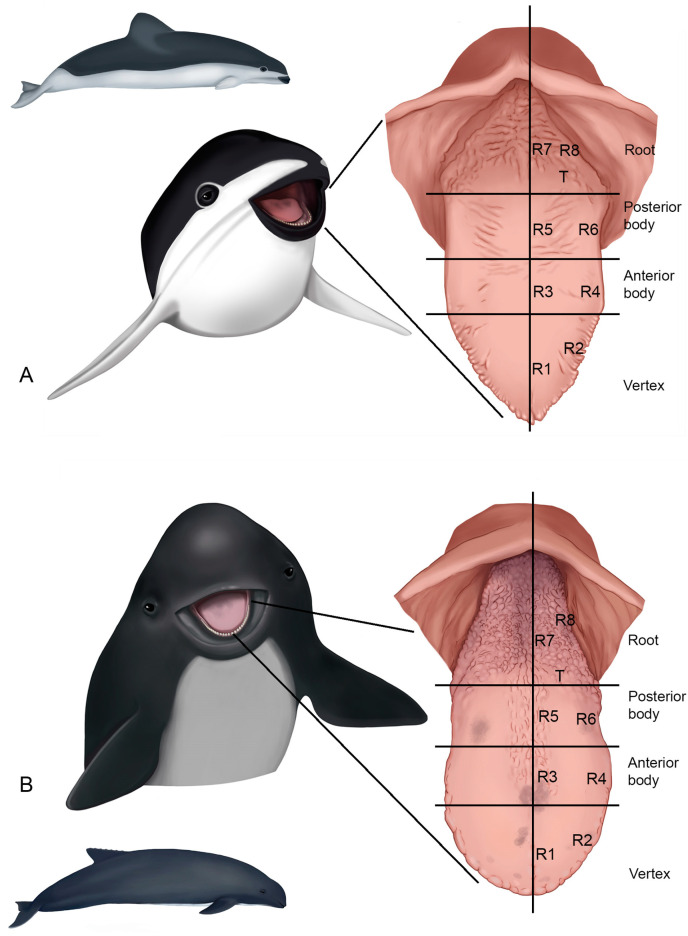
Regions studied in each tongue. (**A**) *Phocoena dioptrica*, (**B**) *Phocoena spinipinnis.* Medial vertex (R1), lateral vertex (R2), medial anterior body (R3), lateral anterior body (R4), medial posterior body (R5), lateral posterior body (R6), transition (T), medial root (R7), and lateral root (R8).

**Figure 2 animals-14-03481-f002:**
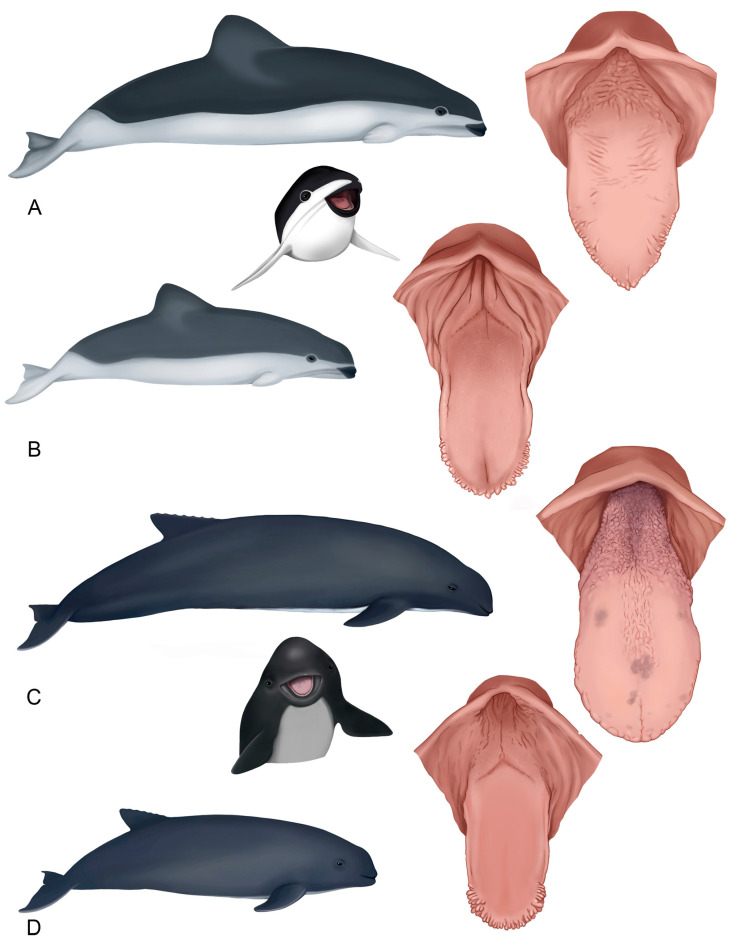
Diagram of the general tongue morphology of both species. (**A**) Adult *P. dioptrica*. (**B**) Juvenile *P. dioptrica*. (**C**) Adult *P. spinipinnis*. (**D**) Juvenile *P. spinipinnis*.

**Figure 3 animals-14-03481-f003:**
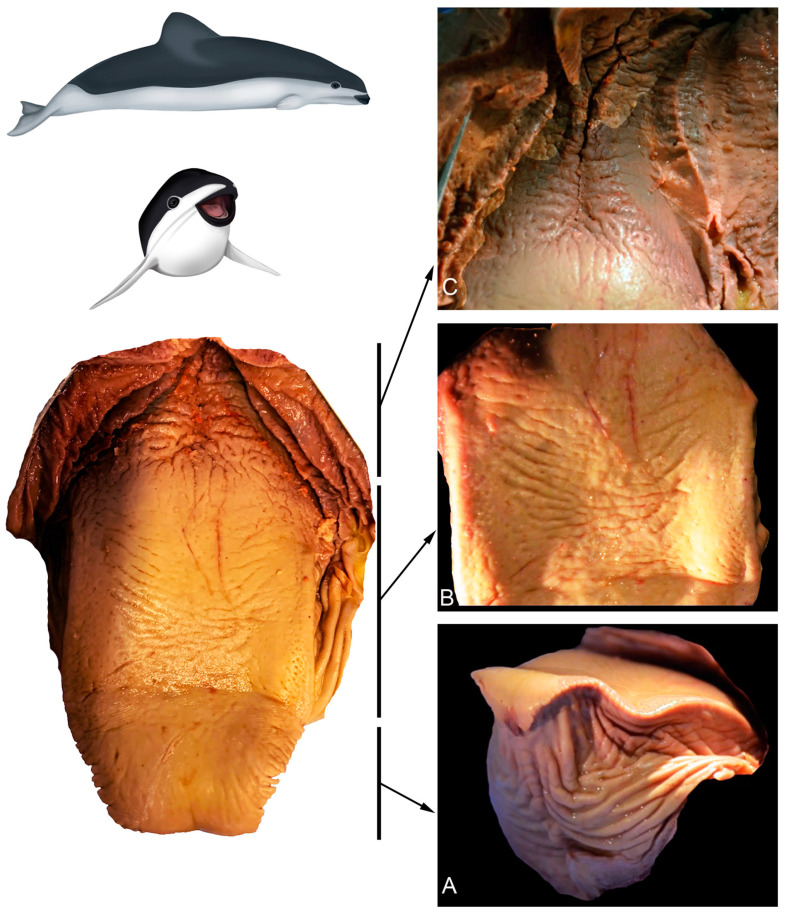
Adult *P. dioptrica*. General tongue morphology. (**A**) Vertex, (**B**) body, and (**C**) root.

**Figure 4 animals-14-03481-f004:**
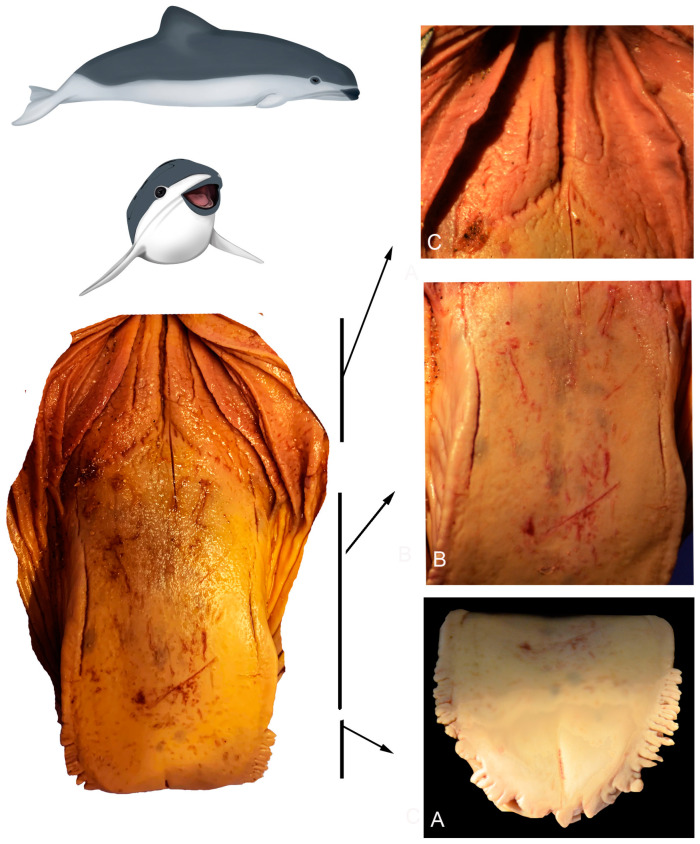
Juvenile *P. dioptrica*. General tongue morphology. (**A**) Vertex, (**B**) body, and (**C**) root.

**Figure 5 animals-14-03481-f005:**
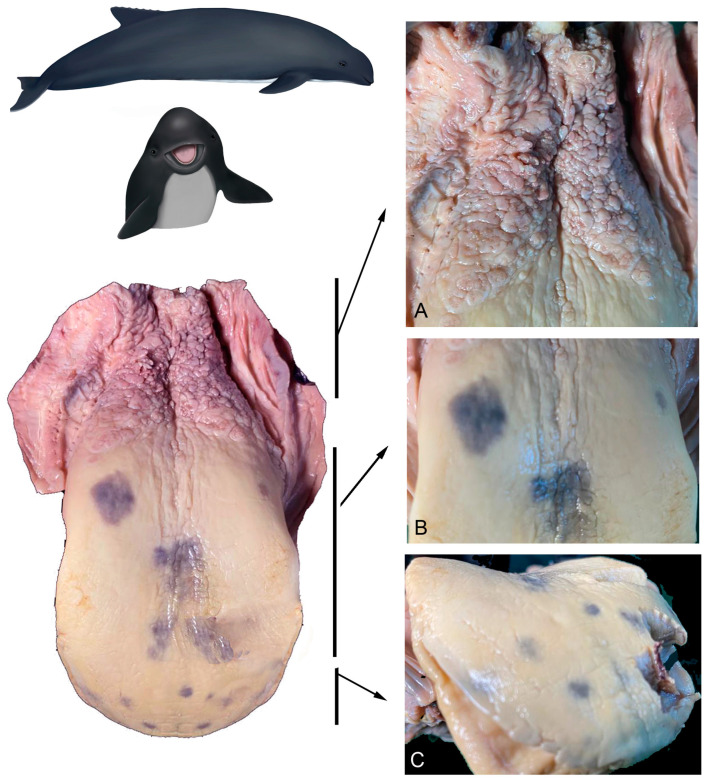
Adult *P. spinipinnis*. General tongue morphology. (**A**) Root, (**B**) body, and (**C**) vertex.

**Figure 6 animals-14-03481-f006:**
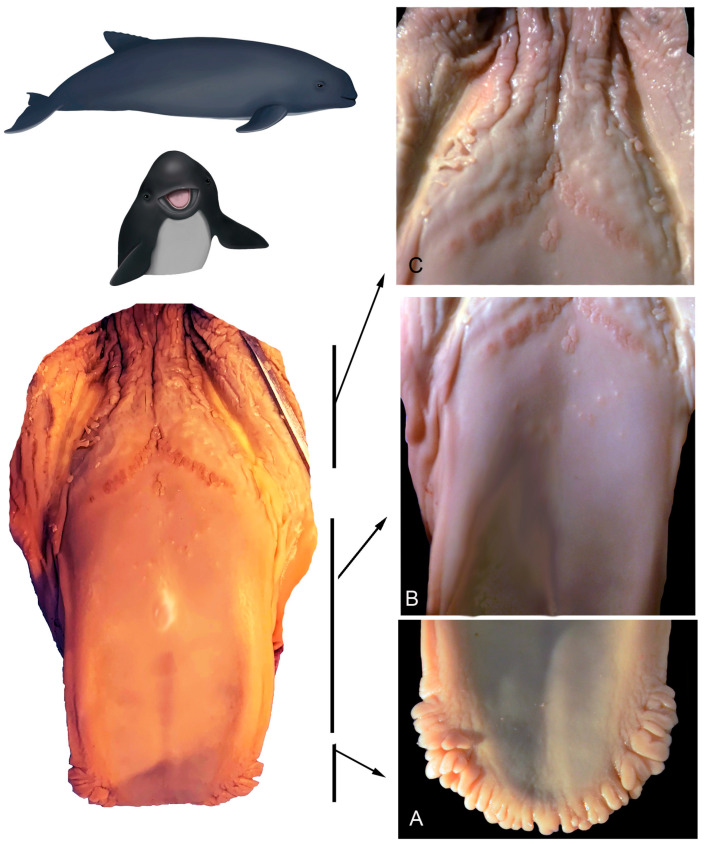
Juvenile *P. spinipinnis*. General tongue morphology. (**A**) Vertex, (**B**) body, and (**C**) root.

**Figure 7 animals-14-03481-f007:**
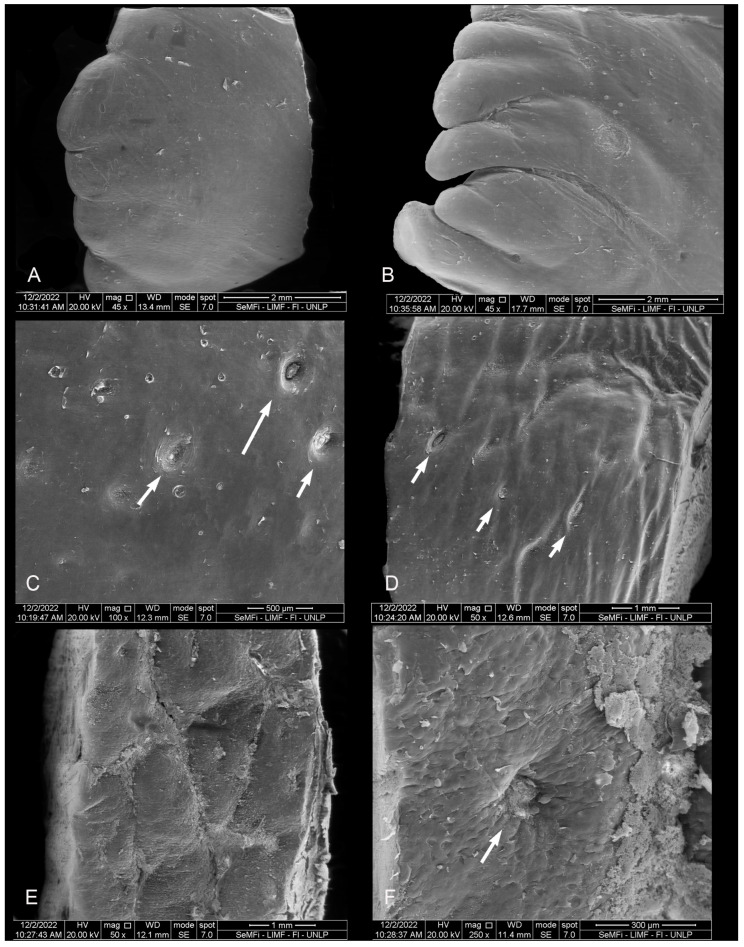
Adult *P. dioptrica*. Surface regions of the tongue observed with scanning electron microscopy: (**A**) anterior vertex, (**B**) posterior vertex, (**C**) anterior body, (**D**) posterior body, (**E**) root, (**F**) details of the pores in the root.

**Figure 8 animals-14-03481-f008:**
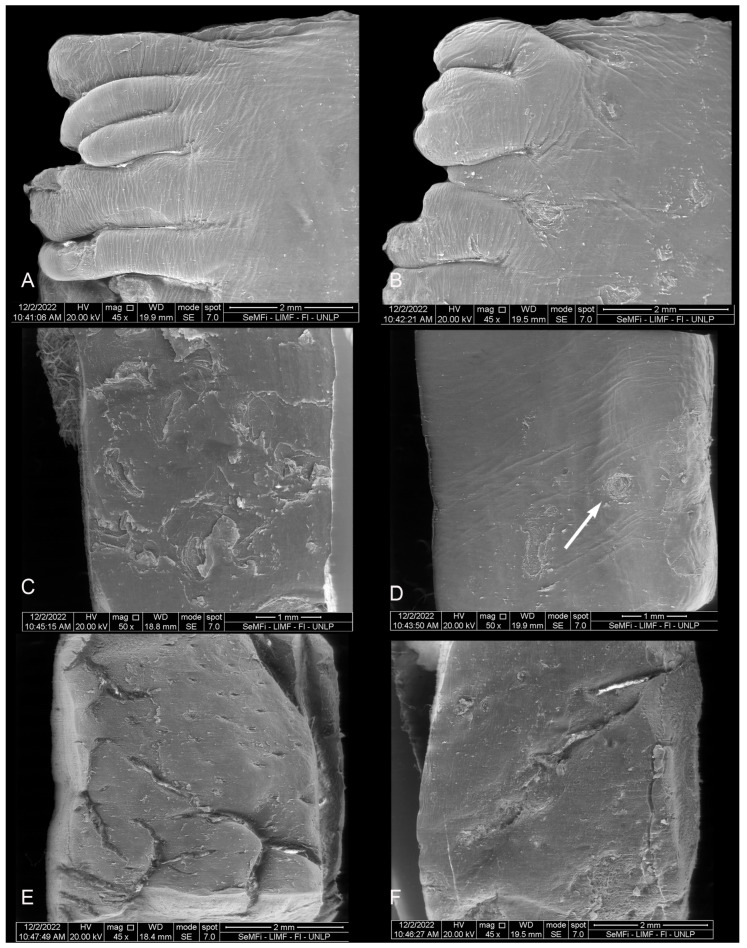
Juvenile *P. dioptrica*. Surface regions of the tongue observed with scanning electron microscopy: (**A**) anterior vertex, (**B**) posterior vertex, (**C**) medial body, (**D**) lateral body, (**E**) root, (**F**) details of the root.

**Figure 9 animals-14-03481-f009:**
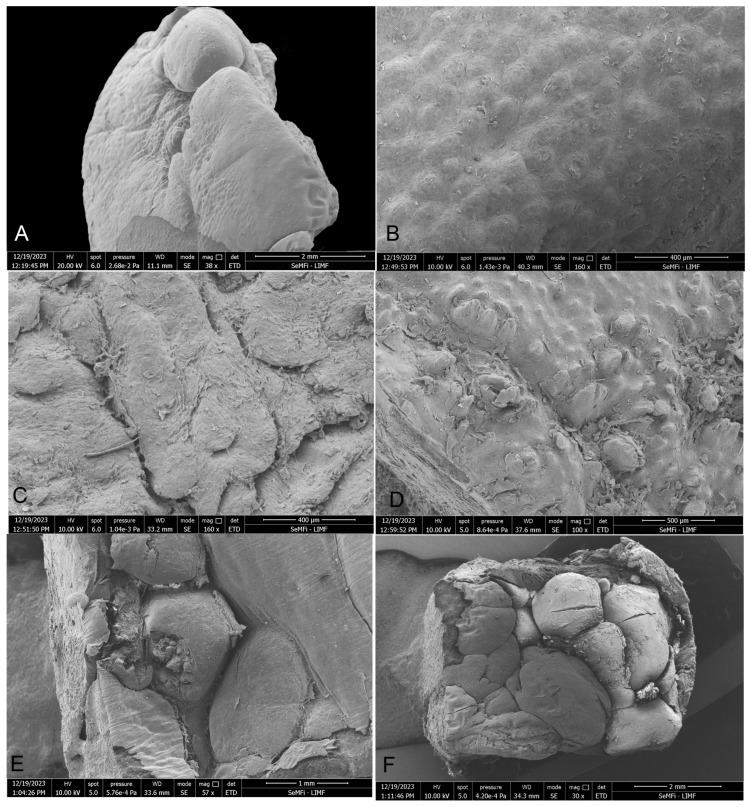
Adult *P. spinipinnis*. Surface regions of the tongue observed with scanning electron microscopy: (**A**) lateral vertex, (**B**) surface of the vertex, (**C**) medial anterior body, (**D**) lateral anterior body, (**E**) root, (**F**) details of the pores in the root.

**Figure 10 animals-14-03481-f010:**
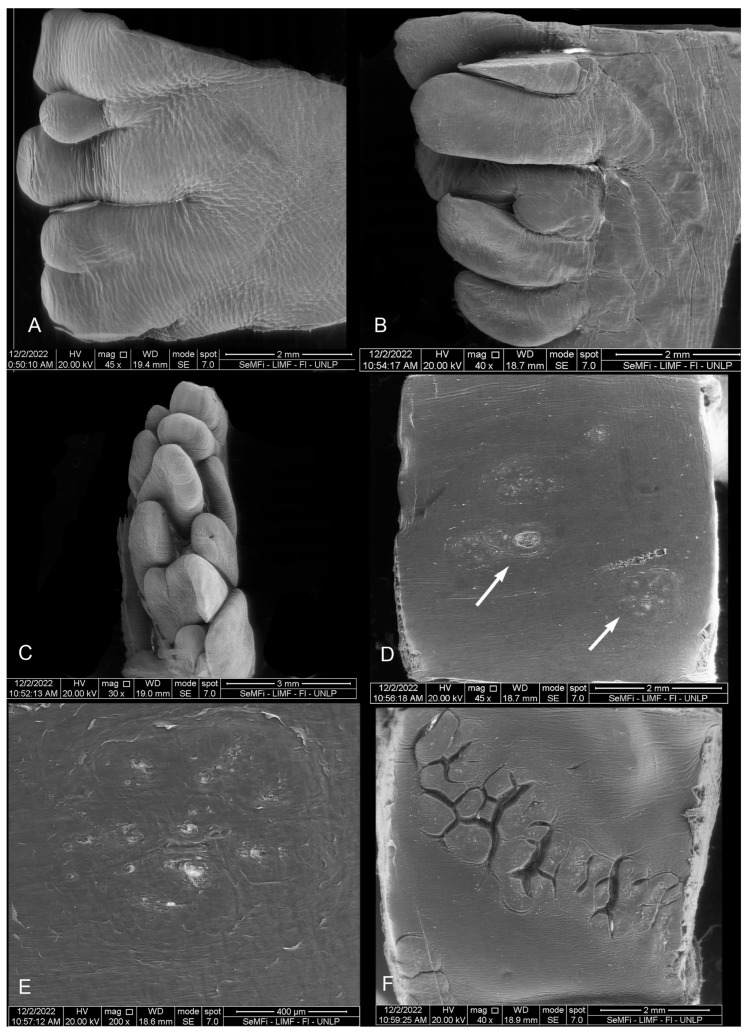
Juvenile *P. spinipinnis*. Surface regions of the tongue observed with scanning electron microscopy: (**A**) anterior vertex, (**B**) posterior vertex, (**C**) rows of marginal papillae, (**D**) anterior body, (**E**) details of the pores in the anterior body, (**F**) crypts in the transition zone.

**Figure 11 animals-14-03481-f011:**
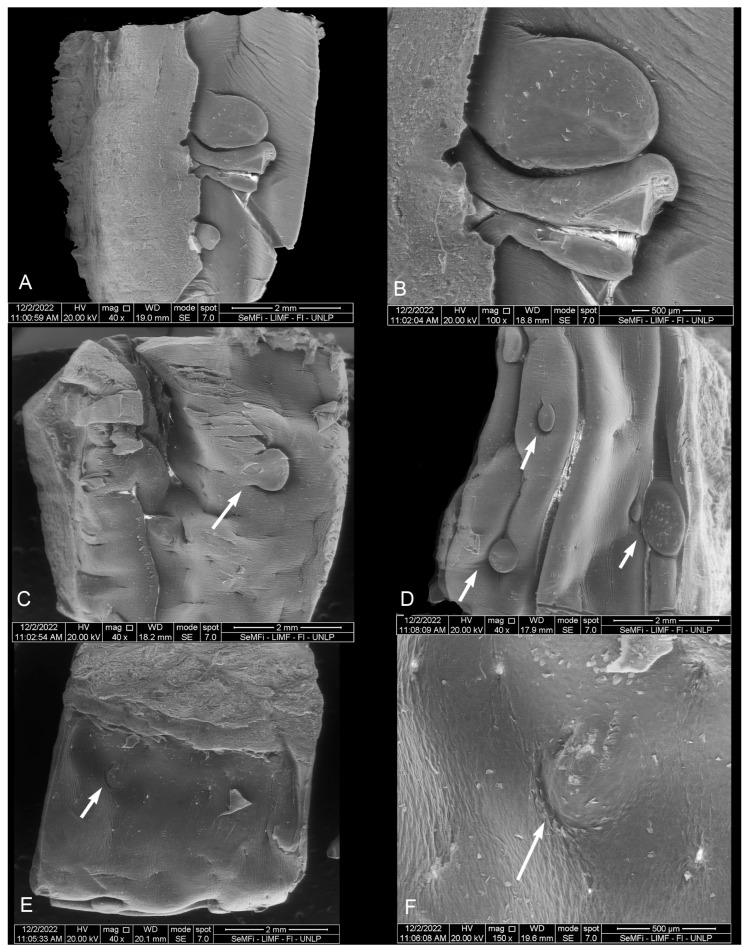
Juvenile *P. spinipinnis*. Surface regions of the root observed with scanning electron microscopy: (**A**) general view of the root, (**B**–**F**) details of the papillae.

**Figure 12 animals-14-03481-f012:**
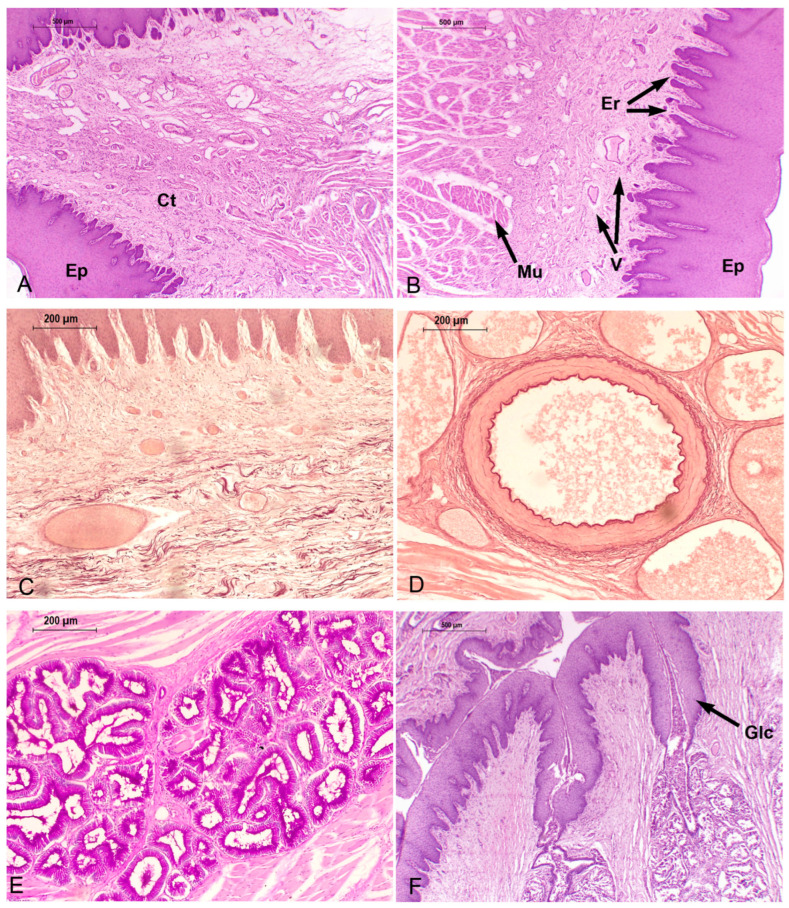
Adult *P. dioptrica*. Microscopical characteristics of tongue regions: (**A**) lateral anterior vertex (4×) H-E, (**B**) posterior vertex (4×) H-E, (**C**) anterior body (10×) Orcein, (**D**) detail of the countercurrent system, orcein (10×), (**E**) PAS-positive glands, in the transition zone (4×) (**F**) root (4×). Ct: connective tissue, Ep: epithelium, Er: epithelial ridge, Glc: glandular duct, Mu: muscle fibers, V: vessels.

**Figure 13 animals-14-03481-f013:**
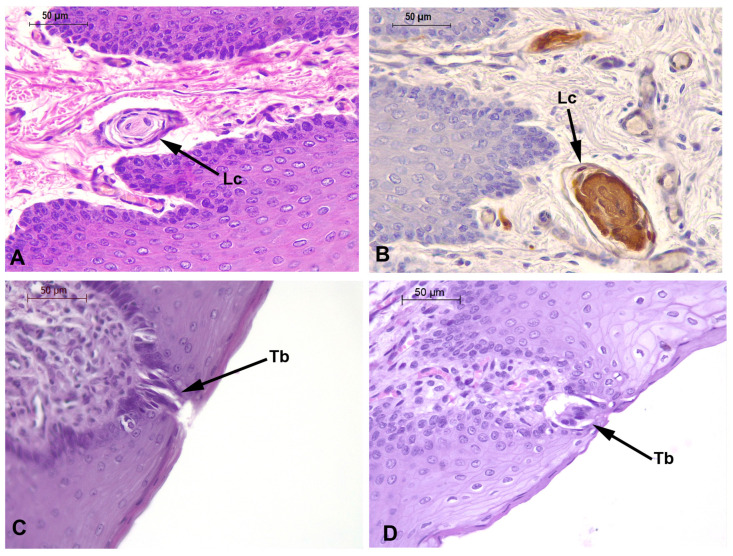
Sensory structures. Lamellar corpuscles in (**A**) *P. dioptrica* (M6021) H-E, (**B**) *P. dioptrica* (M6021). Immunohistochemistry anti-S-100 and taste bud in (**C**) *P. spinipinnis* (M8522), (**D**) *P. spinipinnis* (M12921). Lc: lamellar corpuscle; Tb: taste bud.

**Figure 14 animals-14-03481-f014:**
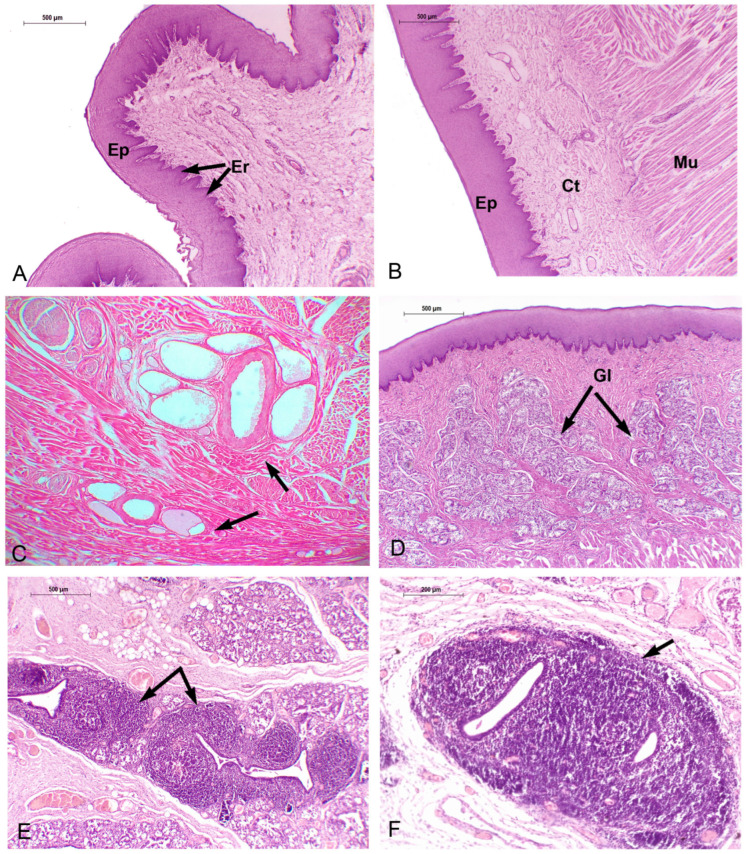
Juvenile *P. dioptrica.* Microscopical characteristics of tongue regions, H-E. (**A**) Lateral anterior vertex (4×), (**B**) anterior body (4×), (**C**) detail of the countercurrent system (black arrows) (4×), (**D**) transition (4×), (**E**) root, associated-lymphoid tissue (black arrows) (4×), (**F**) details of lymphoid tissue (black arrow) (10×). Abbreviations: Ct: connective tissue, Ep: epithelium, Er: epithelial ridge, Gl: glands, Mu: muscle fibers.

**Figure 15 animals-14-03481-f015:**
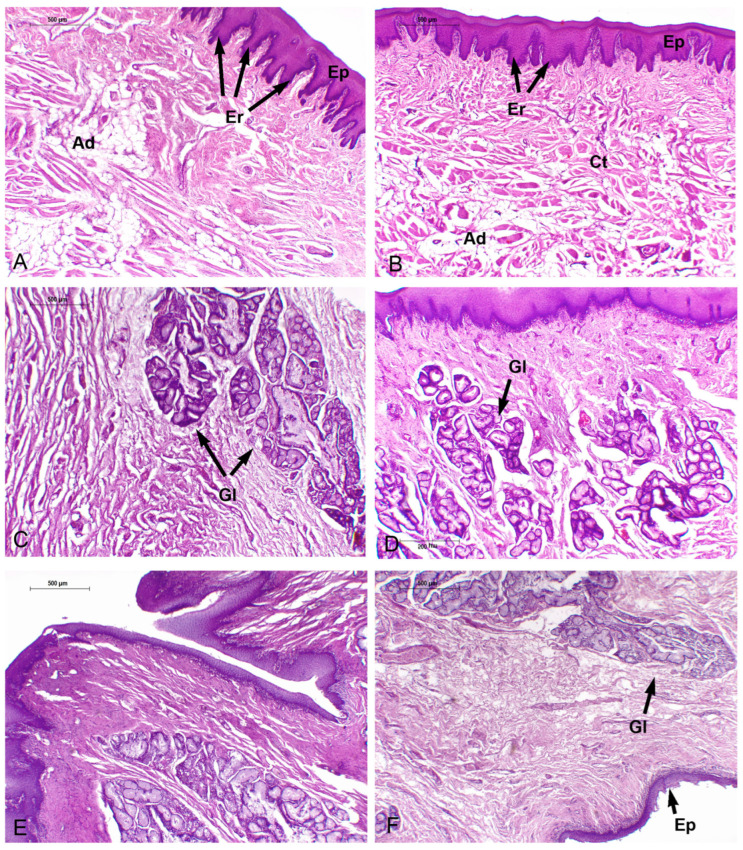
Adult *P. spinipinnis.* Microscopical characteristics of tongue regions, H-E. (**A**) Anterior vertex (4×), (**B**) anterior body (4×), (**C**) posterior body (4×), (**D**) transition (10×), (**E**) anterior root (4×), (**F**) posterior root (4×). Ct: connective tissue, Ep: epithelium, Er: epithelial ridge, Gl: glands, Ad: unilocular adipose tissue.

**Figure 16 animals-14-03481-f016:**
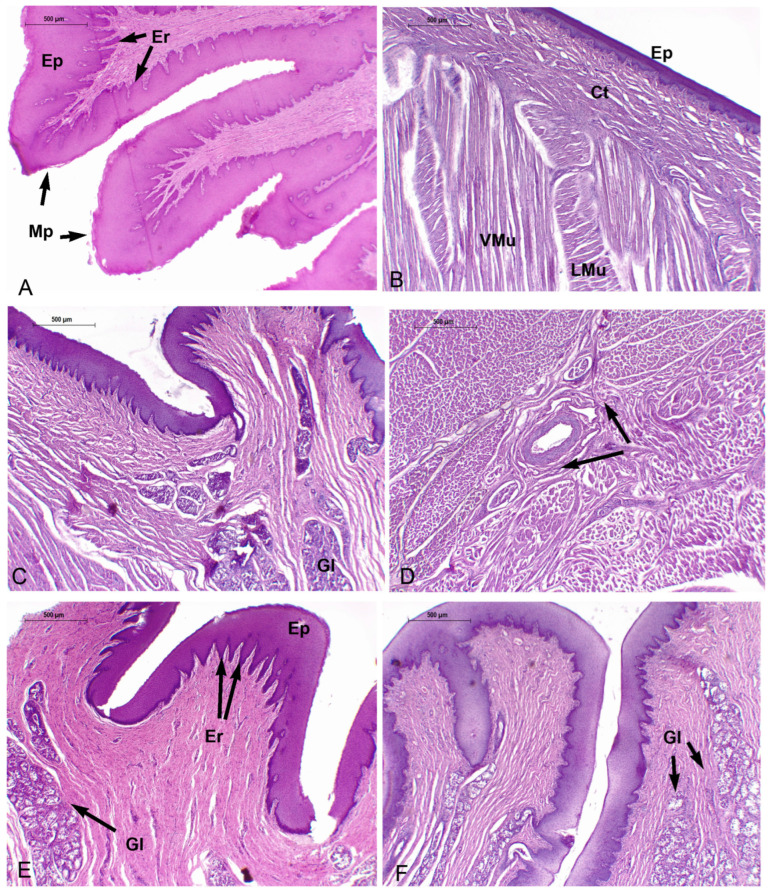
Juvenile *P. spinipinnis*. Microscopical characteristics of tongue regions, H-E. (**A**) Anterior lateral vertex (4×), (**B**) anterior body (4×), (**C**) posterior body (4×), (**D**) countercurrent system (black arrows) (4×), (**E**) transition (4×), (**F**) root (4×). Ct: connective tissue, Ep: epithelium, Er: epithelial ridge, Gl: glands, Vmu: vertical muscle fibers, Lmu: longitudinal muscle fibers, Mp: marginal papillae.

**Figure 17 animals-14-03481-f017:**
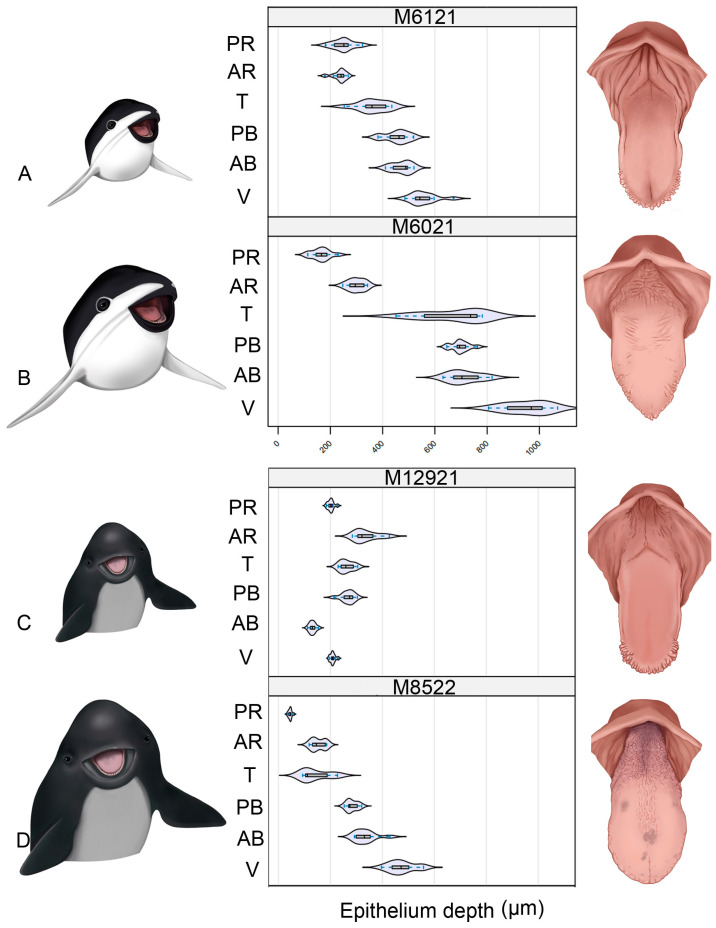
Distribution of tongue epithelial thickness values across each zone and individual. HE. (**A**) Juvenile *P. dioptrica*, (**B**) adult *P. dioptrica*, (**C**) juvenile *P. spinipinnis*, (**D**) adult *P. spinipinnis*. AB: anterior body, AR: anterior root, PB: posterior body, PR: posterior root, T: transition, V: vertex.

**Figure 18 animals-14-03481-f018:**
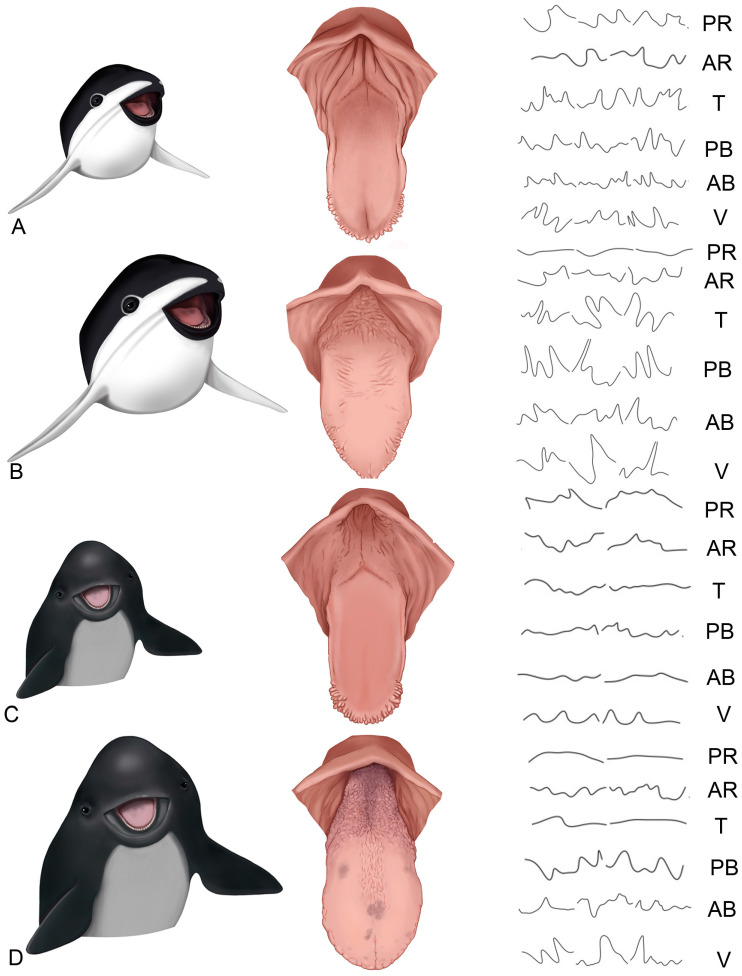
Epithelial ridges, formed between the contact of the epithelium and the underlying connective tissue, exhibit different morphologies and depths. (**A**) Juvenile *P. dioptrica*, (**B**) adult *P. dioptrica*, (**C**) juvenile *P. spinipinnis*, (**D**) adult *P. spinipinnis.* AB: anterior body, AR: anterior root, PB: posterior body, PR: posterior root, T: transition, V: vertex.

**Figure 19 animals-14-03481-f019:**
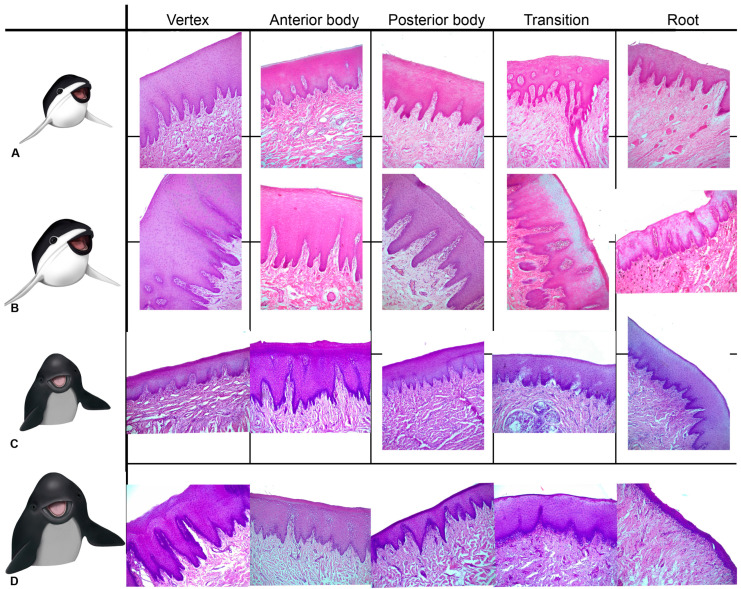
Differences in the epithelial thickness among different lingual regions. HE. (**A**) Juvenile *P. dioptrica*, (**B**) adult *P. dioptrica*, (**C**) juvenile *P. spinipinnis*, (**D**) adult *P. spinipinnis*.

**Figure 20 animals-14-03481-f020:**
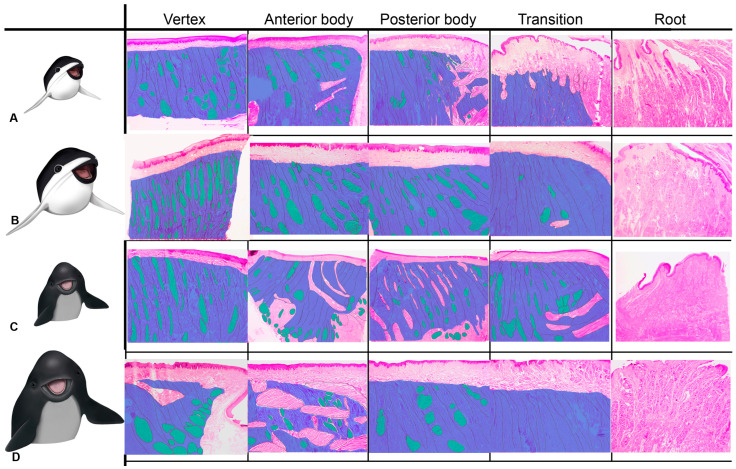
Distribution of different muscle bundles among different lingual regions. Blue: dorsal-ventral (or vertical) fibers; green: antero-posterior (or longitudinal) fibers. (**A**) Juvenile *P. dioptrica*, (**B**) adult *P. dioptrica*, (**C**) juvenile *P. spinipinnis*, (**D**) adult *P. spinipinnis*.

**Figure 21 animals-14-03481-f021:**
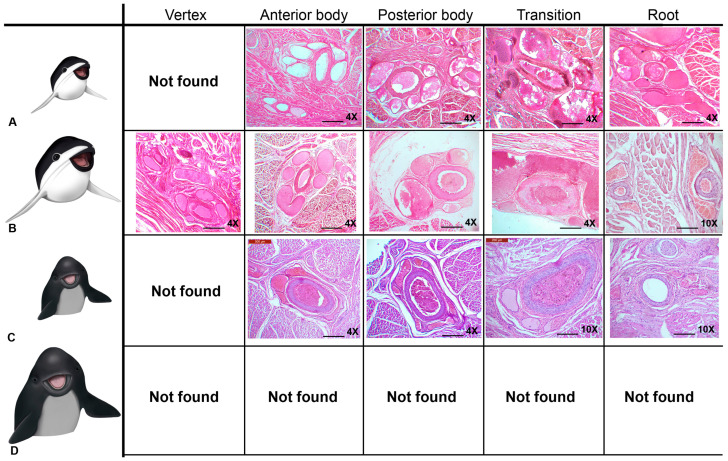
Cross-sectional views of the countercurrent systems in different lingual regions, HE. (**A**) Juvenile *P. dioptrica*, (**B**) adult *P. dioptrica*, (**C**) juvenile *P. spinipinnis*, (**D**) adult *P. spinipinnis.* Scale bar: 500 µm at 4×, and 200 µm at 10×.

**Figure 22 animals-14-03481-f022:**
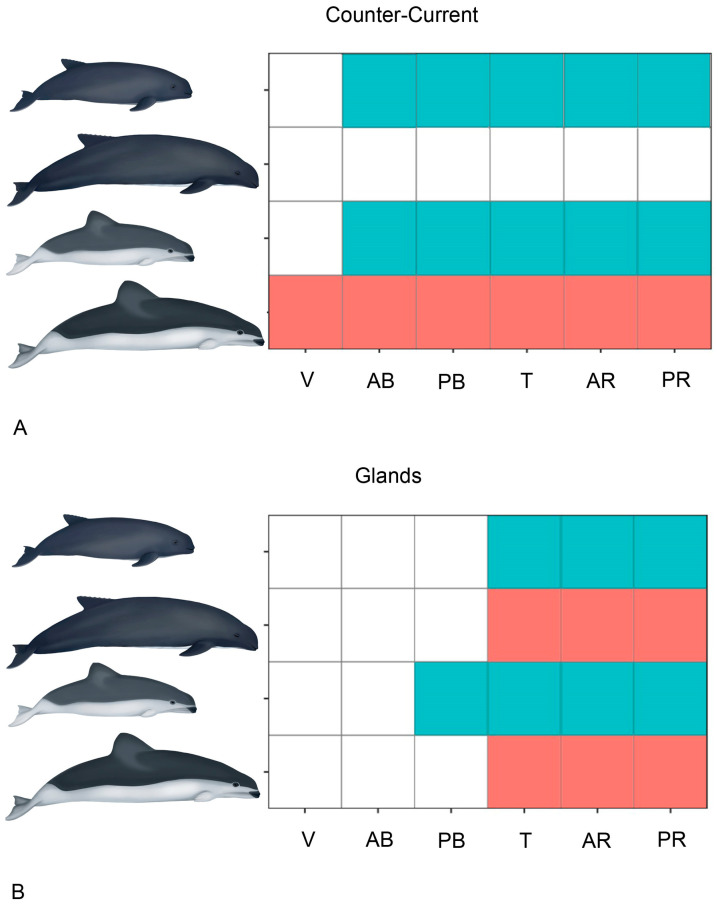
Comparative graphs representing the presence of the countercurrent system and glands across at different regions and each species. (**A**) *Phocoena dioptrica*, (**B**) *Phocoena spinipinnis*. V—vertex, AB—anterior body, PB—posterior body, T—transition, AR—anterior root, PR—posterior root. Turquoise = juvenile. Deep rose = adults.

## Data Availability

All data generated or analyzed during this study are included in this published paper.

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
