# Peer review of "Ecomorphological and Age-Related Adaptations in the Tongues of Phocoena dioptrica (Spectacled Porpoise) and Phocoena spinipinnis (Burmeister’s Porpoise) (Phocoenidae: Cetacea)"

_animals, 2024, doi:10.3390/ani14233481_

Round 1
Reviewer 1 Report
Comments and Suggestions for Authors
Eco-morphological and age-related adaptations in the tongues of Phocoena dioptrica (Spectacled Porpoise) and Phocoena spinipinnis (Burmeister´s Porpoise) (Phocoenidae: Cetacea)
Abstract
“The vertebrates tongue´s reflect part of its adaptations to diverse feeding strategies, types of food items they eat and the environments where they live. . Our contribution was to analyzethe macro- and microscopic morphology of the tongue of two porpoise species (Phocoena dioptrica and Phocoena spinipinnis, juveniles and adults). Macroscopic and microscopic studies (conventional histology, scanning electron microscopy im munohistochemistry and morphometry) were performed. Differences between juvenile and adult individuals of the same species, as well as between juveniles and adults of both species, were found, probably related to their feeding and/or geographical distribution. In addition, novel aspects related to ontogenetic morphological differences, thermoregulation, and immune system components were described. We found a lingual countercurrent vascular system (periarterial venous retia), only mentioned for mysticetes and Physeter macrocephalus (never for smaller odontocetes), In addition, we identified mechanorreceptors (lamellar corpuscules). Both species showed marginal papillae, but only in P. spinipinnis small (probably vestigial), taste buds were observed. Finally, lingual lymphoid aggregates were found The abstract is too long and provides speculative approaches that are not needed here
Introduction
“The tongue has been studied in numerous taxa of terrestrial mammals, particularly in domestic species [3], but also in other groups such as armadillos and non-domestic carnivores [4,5,6,7, 8].”
Please revise this sentence. A lot of studies have been made on the tongue morphology of a large number of mammals in the last three years. Some examples:
Morphological structure of the tongue of the European badger (Meles meles). A Haligur, S Ozkadif, A Alan, M Haligur - Folia Morphologica, 2022 - journals.viamedica.pl
Comparative evaluation of the ultrastructural morphology and distribution of filiform and fungiform tongue papillae in Egyptian mice, fruit bats and long-eared …T Haggag, EF Mahmoud, ZA Salem… - Anatomy & Cell …, 2020 - synapse.koreamed.org
Comparative Study of Lingual Papillae, Lingual Glands and Lyssa of the Tongue of Selected Wild Felids (Carnivora, Felidae) in Biological Aspects K Goździewska-HarÅ‚ajczuk, K Barszcz… - Biology, 2023 - mdpi.com
Microscopic structure of the tongue in the lesser hedgehog tenrec (Echinops telfairi, Afrosoricida) and its relation to phylogenesis. P Cizek, P Hamouzova… - Anatomical Science …, 2020 - Springer
“Considering the relevance of the tongue, and that little is known about that organ on Phocoena dioptrica and Phocoena spinipinnis, the objective of our contribution is to describe the macro- and microscopic morphology of the tongue of that two porpoise… for which little is known about their biology”
L. 81… What is the objective of the paragraph? Phylogenetic context? It is not presented in this paper.
You should rewrite the introduction (phylogeny?) objective in the context of evolution (phylogeny?) and ecology (feeding strategies? Diet?). The Comparison is interesting but why?
“Regarding the immune system in cetaceans…”: the paragraph can be shortened
Material and Methods
Clear and simple
Results
Very descriptive. But nice figures. Clear description of all the lingual features.
Discussion
A table with comparative data in marine tetrapods should be useful and enough for a short discussion. You should discuss the points that can be compared with other species i.e., cetaceans and others. Reduce the discussion. Several paragraphs can be reduced. The results are repeated in the discussion. Only figures from the results can be indicated to help the reader understand the points in the discussion.
“…might be associated with its diet”: Explain why?
L. 578-621: The question of the milk can be shortened. Very speculative
L. 635: not useful discussion. Again, it is too long to say: “nothing is sure”. Reduce this section to one or two lines.
L. 644: same this paragraph is useless.
Again the conclusion rewrites the results and the discussion:
The general morphology of the tongues of the two porpoise species here studied aligns with that of other cetaceans previously analyzed. However, we document a range of novel aspects related to ontogenetic morphological differences, thermoregulation structures and the immune system components. A lingual countercurrent vascular system is identified, Particularly, in P. Spinipinnis small and likely vestigial taste buds were observed, which were not associated with any evident papilla or serous glands. Finally, lingual lymphoid aggregateshad not been previously reported in cetaceans.
Conclusion: this interesting and descriptive paper deserves to be published. But the discussion must be shortened.
Author Response
"Please see the attachment."

Reviewer 2 Report
Comments and Suggestions for Authors
Could the authors explained how they defined juveniles over adults? based on tooth aging, size or reproductive tract immature vs mature? Could they include the rational in the method part?
Few comments regarding typos: I suggest the authors review their manuscript to screen through.
Few typos/space errors to be aware of:
- In contrast, juvenil P. spinipinnis shows a thicker epithelium than that of the adult 539
P. 662 spinipinnis. Need to be in Italic
Physeter macrocephalus (but not for smaller 766. Italic please
Their identification only in the juvenil 700 specimen of P. spinipinnis could be understood
exposure.Although macroscopic and microscopic descriptions of lym-728
waters.Given the limited knowledge about the 703
large space: ported for the laryngeal and anal tonsils of 749
T. truncatus [52].
The general morphology of the tongues of the two porpoise species here studied 757: Isuggest species studied in this paper.
The sentence is incomplete or makes no sense: the most prominent ridges those present in the adult (Figure 18). The areas of great-554
Why is the conclusion formatted differently than 1,2,3 and 4.? Is it normal?
Figure 22.why using blue and red. Could you describe the color coding in the legend.
About taste buds: It would bring additional insights to read the following paper and to include the role of taste buds in immune response in the discussion. i suggest adding this reference and to include it appropriately : Barboza, Meghan Lee Bills, and Beau Reyno. "Taste receptors in aquatic mammals: Potential role of solitary chemosensory cells in immune responses." The Anatomical Record 305, no. 3 (2022): 680-687.
Author Response
"Please see the attachment."

Round 2
Reviewer 1 Report
Comments and Suggestions for Authors
Nice paper. Hope that it will be used as a reference for tongue morphology.
Line 776: appreciate